# Enhancing the Production of Syngas from Spent Green Tea Waste through Dual-Stage Pyrolysis and Catalytic Cracking

Asma Ben Abdallah [1,2], Aïda Ben Hassen Trabelsi [2], Alberto Veses [3,*], Tomás García [3], José Manuel López [3], María Victoria Navarro [3] and Daoued Mihoubi [2]

1   Department of Energy Engineering, National School of Engineers of Monastir, University of Monastir, Monastir 5000, Tunisia; benabdallahasma1993@gmail.com
2   Laboratory of Wind Energy Management and Waste Energy Recovery (LMEEVED), Research and Technology Center of Energy (CRTEn), B.P. 95, Hammam-Lif 2050, Tunisia; aida.benhassen@crten.rnrt.tn (A.B.H.T.); daoued.mihoubi@crten.rnrt.tn (D.M.)
3   Instituto de Carboquímica (ICB-CSIC), C/Miguel Luesma Castán 4, 50018 Zaragoza, Spain; tomas@icb.csic.es (T.G.); jmlopez@icb.csic.es (J.M.L.); navarro@icb.csic.es (M.V.N.)
*   Correspondence: a.veses@icb.csic.es

**Abstract:** A sequential two-step thermochemical process was studied for spent green tea waste (SGTW), involving an initial pyrolysis step followed by thermal or catalytic cracking. This process was carried out in two bench-scale reactors (fixed bed reactor and tubular reactor) serially coupled. At a fixed pyrolysis temperature of SGTW (550 °C), the application of high cracking temperatures (700 and 800 °C) positively affected both the yield and composition of the gas product. Consequently, it has the potential to be used for the production of diverse biofuels and chemicals, or to be partially recycled to optimize the process efficiency. Moreover, the use of inexpensive catalysts, particularly dolomite, was considered advantageous, since the syngas yield (56.5 wt%) and its potential were greatly enhanced, reaching a $H_2/CO$ ratio of 1.5. The homogenous biochar obtained, with a calorific value of 26.84 MJ/kg, could be harnessed as good-quality fuel for briquette applications and as a biofuel source for generating stationary power. Furthermore, catalytic cracking pyrolysis was examined for different types of coffee waste, revealing that this process is a simple and clean solution to valorize oxygen-rich lignocellulosic biomass and generate valuable gaseous by-products.

**Keywords:** spent green tea waste; biomass pyrolysis; catalyst; dolomite; syngas





## 1. Introduction

Fossil fuels such as oil, coal, and natural gas have been significant to human development since they could be burned to supply heat for direct use (e.g., for heating or cooking), to produce electricity, or to power engines (e.g., the internal combustion engines in cars and buses). However, rising fossil fuel prices and energy demand, significant concerns about supply security, and environmental crises are the main drivers for the search for various alternative sources of renewable energy. Being an abundant, inexpensive, environmentally friendly, and renewable energy source, biomass stands as one of the primary alternatives to replace fossil fuels, thus contributing to the decarbonization of both the energy and chemical sectors. Biomass resources cover a wide range of materials, such as energy crops, organic wastes, forest residues, urban wastes, and agro-industrial residues. Within agro-industrial waste, spent green tea is an example of readily available and largely underutilized biomass waste. The great popularity of the tea drink led to a worldwide increase in its consumption each year, reaching 6.6 million tons in 2021 [1]. Particularly, the consumption of tea-based beverages generates 90% of tea waste [2]. Spent green tea waste (SGTW), as a lignocellulosic biomass, is composed of lignin, hemicellulose, and cellulose and has a remarkable energy content, which could lead to the generation of

value-added products through thermochemical conversion processes. Thermochemical processes are considered as the most promising practice of biomass waste treatment compared to biological conversion technology, as they provide pollution control and high-energy recovery. Among the different processes available for the thermochemical conversion of biomass wastes, pyrolysis is an advantageous process and an important method to efficiently convert the organic material into high-energy valuable biofuels with less cost and few environmental problems in comparison with gasification and combustion processes [3]. Biofuels have been popular as alternative vehicle fuels for a while. However, they also have various uses such as for lubrication, cleaning oil spills and grease, removing paint and adhesive, charging electronics, heating, energy generation, and transportation (e.g., airplane and marine engines). In recent years, the rising costs of energy and raw materials (such as vegetable) have mainly been responsible for the higher biofuel (e.g., biodiesel) prices. For these reasons, according to the International Energy Agency (IEA), a USD 1 trillion investment in low-carbon energy (such as biomass energy) is needed by 2030 to avoid catastrophic climate change effects [4].

The biomass pyrolysis process is carried out in a non-oxidizing atmosphere and at moderate temperatures (300–600 °C). Three fractions result from this process: a carbon-rich solid (biochar), non-condensable gases (pyrogas), and liquid products (bio-oil). These potential products can be used in various applications or be partially recycled to meet the energy needs of the pyrolysis process. The distribution and properties of the pyrolysis products are strongly influenced by various factors, namely the type and configuration of the reactor; the operating conditions such as temperature, volatile residence time, and heating rate; biomass sources; and even the harvesting time or the growing environment for the same waste. Tea waste is usually disposed of in landfills or digested to biogas through anaerobic digestion [5], and only some studies have been conducted to examine the potential of tea waste to produce valuable products through the conventional pyrolysis process [6,7]. Based on these tests, it was observed that low mass yields were obtained with low energy yields of bio-oil, which exhibited a high moisture content (>50 wt%) [8]. Thus, the direct use of this bio-oil is quite limited. Moreover, the poor quality of bio-oil derived from tea waste pyrolysis was confirmed by the presence of unwanted oxygenated compounds (such as carboxylic acids, hydroxyketones, hydroxyaldehydes, and phenols), which resulted in lower calorific values (10.1 MJ/kg at 550 °C) [8] compared to other common lignocellulosic biomass pyrolysis oils [9]. Most of these works have been performed in one-step conventional pyrolysis at different ranges of temperatures. Although this single-step pyrolysis process could be advantageous from an economical point of view, pyrolysis combined with a further cracking step could be proposed as an interesting alternative in order to convert that poor-quality oil fraction into a high-quality gas stream.

Catalysts are extensively applied in the pyrolysis process of biomass and other residues due to its important role and significant impact on the distribution and properties of pyrolysis products. Generally, the use of catalysts improves the kinetics reaction of pyrolysis by breaking down higher-molecular-weight compounds into lighter hydrocarbon products [10]. Based on the composition and type of catalysts, they can be divided into two categories: natural and synthetic metal catalysts such as $Na_2CO_3$, $K_2CO_3$, $ZnCl_2$ [10], $CeO_2$, Rh, $SiO_2$ [11], Ni, CeO2, $Al_2O_3$ [12], and different types of zeolites [13]. Chen et al. [14] investigated the effect of adding 2% of metal oxides, magnesium sulphate, and chlorides on the pyrolysis of aspen pellets. The results showed that metal oxides improved heavy oil yield, whereas magnesium sulphate and chlorides favored the production of water phase residue. However, nitrates favored the syngas production, which is mainly composed of hydrogen (25%), carbon monoxide (44%), and methane (18%). Qu et al. [15] explored the influence of using Ni/Fe bimetal ZSM-5 as a catalyst on the catalytic pyrolysis kinetics and the product analysis of waste tires and found that ZSM-5 loading with 7 wt% Ni and 3 wt% Fe reduced the activation energy by 13%; thus, the best catalytic effect was achieved. Moreover, analysis results indicated that metallic-Ni-based catalysts were effective at converting the alkenes into aromatic hydrocarbons. Huang et al. [16] examined the effect of particle

size of $Al_2O_3$ (10-mesh and 50-mesh) on the microwave pyrolysis products of corn stover. This study found that the addition of 10-mesh $Al_2O_3$ to the feedstock reduced the liquid yield, while the gas yield increased. These results could be explained by the fact that small catalyst particles are encapsulated by biomass particles and other catalysts, reducing their catalytic activity.

Although the yield and product quality from catalytic pyrolysis depend on several variables such as biomass and catalyst type or operating conditions such as temperature, catalyst-to-biomass ratio, reactor type, or vapor residence time, various metal-based catalysts, particularly alkali-metal- and Ni-based catalysts, have proven to be effective in bio-oil upgrading and in the removal of heavy tar, achieving tar removal rates exceeding 99%. However, they can become inactive over several cycles due to the carbon deposition [17]. Compared to synthetic metal catalysts, the natural catalysts (e.g., limestone, bentonite clay, red mud, sepiolite, calcite, dolomite, olivine) are more cost-effective and readily available, making them preferable for use. From the literature, it was observed that calcined calcite and dolomite were efficient in the deoxygenation process and also in the gas distribution, thus promoting $CO_2$ capture due to the presence of CaO. Other catalysts, such as olivine, exhibit a similar behavior to dolomite, effectively converting tar. While most studies are focused on bio-oil upgrading, where more deoxygenated bio-oil can be obtained, especially from specific biomass sources like forestry and agricultural residues (e.g., pine woodchips, almond shells, or grape seeds [18–21]), other biomass types with different hemicellulose, cellulose, lignin, and ash compositions can significantly influence product quality. Consequently, conventional pyrolysis may result in low-quality bio-oil. Hence, the search for specific upgrading processes that can be integrated into future biorefineries becomes essential for sustainable processes. In these cases, considering an approach that prioritizes the production of improved and higher-value products such as gas and/or char emerges as a promising and underexplored alternative. For that, the development of a catalytic two-stage process with low-cost catalysts that deals with the problem of the low-quality bio-oil and, at the same time, that keeps the char properties unmodified for further requirements raises a promising strategy for future implementation. This approach widens the possibilities for recovering various types of biomass, including tea or coffee waste, that may not yield high-value bio-oil through conventional pyrolysis methods.

The main purpose of this present work is to turn the wheel from waste to wealth via sustainable spent green tea management applications over landfill disposition. For this reason, both non-catalytic and catalytic cracking pyrolysis processes were performed to valorize the organic contents of SGTW. Experiments were carried out inside a novel configuration, aiming to avoid unwanted by-products such as tar whilst producing potential high-quality syngas and biochar. This two-stage technology consists of two reactors heated in series: a fixed bed (for pyrolysis) and a tubular reactor (for catalytic and cracking processes). Additionally, the influence of cheap and ecofriendly materials, which were incorporated into the cracking step as ex situ catalysts (calcined olivine and calcined dolomite), was assessed. The resulting syngas and biochar were analyzed, and their energetic uses as potential and valuable sources of biofuels were well investigated. To further study the application of the process, the use of regenerated catalysts was also investigated. Finally, the reliability and efficiency of the catalytic cracking pyrolysis process to produce worthy gaseous by-products were also evaluated with different lignocellulosic biomass such as coffee wastes.

## 2. Results and Discussion

### 2.1. Characterization of SGTW

The physicochemical properties of spent green tea are summarized in Table 1. It is worth highlighting the significant VM content (70.25 wt.%), resulting in a higher VM/FC ratio (3.75) compared to other biomasses and coals [22–24]. Therefore, this biomass waste is very reactive and a suitable solid feedstock for the devotalization processes. Moreover, SGTW could easily ignite in thermal systems, such as boilers, if used as solid fuel [25].

SGTW is also characterized by low ash content (3.75 wt.%), with a heterogeneous nature pointed out by the occurrence of different trace inorganic compounds. According to the ultimate analysis of SGTW, it ought to be featured that the carbon content (46.12 wt.%) can be considered similar to that found in other types of lignocellulosic biomass, which implies a high energy content. In addition, high H/C and low O/C ratios indicate significant HHV (19 MJ/kg). Thus, the studied SGTW has great potential for generating clean energy compared to solid fossil fuels. The amount of nitrogen component is noted to be large in comparison with woody biomasses (>1%), which is known to be present in compounds such as caffeine, indole, or amines [8].

**Table 1.** Properties of SGTW.

| Proximate Analysis | Values (wt.%) | Compositional Analysis | Values (wt.%) |
|---|---|---|---|
| Moisture | 7.24 | Extractives | 11.7 |
| Ash | 3.75 | Hemicellulose | 34.0 |
| VM | 70.25 | Cellulose | 21.5 |
| FC * | 18.76 | Lignin | 32.7 |
| VM/FC | 3.75 | | |
| **Ultimate analysis** | **Values (wt.%)** | **Inorganic compound** | **Values (wt.%)** |
| C | 46.12 | Al | 4.0 |
| H | 6.47 | Ca | 21.1 |
| N | 2.79 | Fe | 0.9 |
| S | 0.12 | K | 7.1 |
| O * | 44.5 | Mg | 8.6 |
| H/C | 1.7 | Mn | 2.6 |
| O/C | 0.7 | Na | 5.1 |
| HHV (MJ/kg) | 19.00 | P | 3.8 |
| | | Si | 4.1 |

* Calculated by difference.

## 2.2. Thermal Degradation Behavior

Thermogravimetric analysis (mass loss + rate of mass loss) is a highly beneficial technique to comprehend and study the pyrolysis behavior of SGTW under specific conditions. Figure 1 depicts the curves of TGA and DTG for SGTW. The thermal decomposition process can be split into three pyrolytic stages. The first stage (<180 °C) corresponds to the decomposition of the weakly bonded $H_2O$. The second stage, demarcated within the temperature range of 180–530 °C, is in line with the devolatilization of biomass constituents [26]. A significant drop in the mass loss of SGTW (66.2%) is observed during this stage, indicating the release of a great number of volatile components [27]. Moreover, two significant peaks are noticed at 230 °C and 348 °C, linked to the thermal decomposition of extractives, and hemicellulose and cellulose, respectively. A shoulder is also observed in this stage at 470 °C related to the main endothermic decomposition step of lignin. The third stage (>530 °C) is attributed to the passive pyrolysis process [28]. SGTW mass loss at this third stage (5.6%) is much lower than at the second stage due to the recalcitrant nature of this remaining lignin, which is decomposed in the case of a lack of significant mass conversion reactions. It must be highlighted from the thermogravimetric results that 550 °C is a suitable temperature to ensure an entire devolatilization of SGTW.

The DTG curve is deconvoluted [29] in order to determine the biomass waste composition of lignocellulosic constituents that is compiled in Table 1: 11.7% extractives, 34% hemicellulose, 21.5% cellulose, and 32.7% lignin.

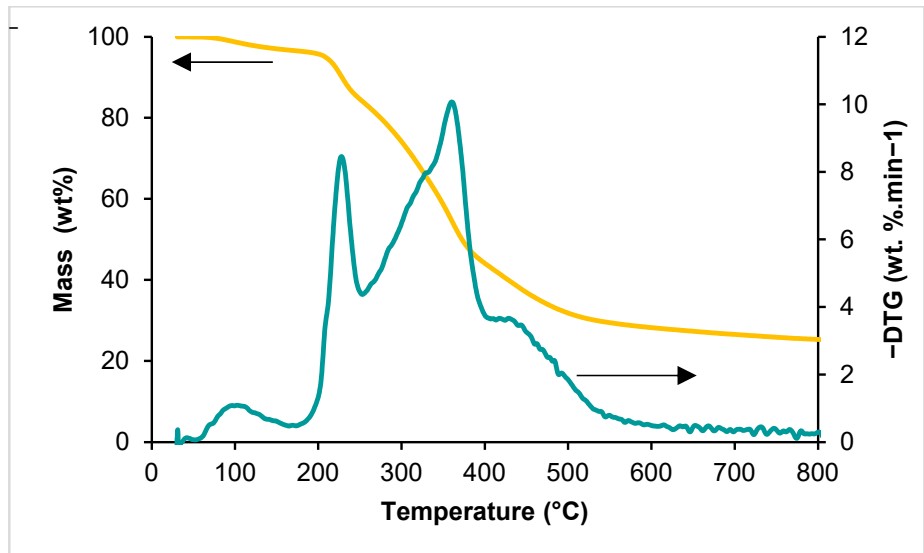

**Figure 1.** TG and DTG curves of SGTW at 25 °C/min.

## 2.3. Process Performances

### 2.3.1. Distribution of Products

The distribution of products, which consisted of bio-oil, biochar, and non-condensable gas, obtained after different experiments carried out in the two-stage process is depicted in Figure 2. The cracking pyrolysis step of SGTW was performed at different temperatures (400, 700, and 800 °C) while keeping the temperature of the pyrolysis step at 550 °C. The bio-oil produced during SGTW pyrolysis experiments was composed of two fractions: an aqueous fraction (pyrolytic water) and an organic fraction (tar), which were separated through centrifugation for 1 h at 1500 rpm (the upper layer represented the pyrolytic water and the bottom layer was the organic fraction). Increasing the cracking temperatures promoted the cracking of tar due to various reactions (e.g., decarbonylation, decarboxylation, polymerization, cyclization, and aromatization reactions) and the dehydrogenation of pyrolytic water. Simultaneously with the decline in tar yield, the non-condensable gas sharply increased from to 29.3 wt.% to 49.8 wt.%. It should be highlighted that when the production of the gas stream was maximized (at 800 °C), the minimum tar fraction was obtained. Accordingly, the higher the cracking temperature, the higher the non-condensable gas production, and the lower the process operational issues, which could occur because of the poor quality of tar. Similar results were already stated by different researchers in previous studies using municipal solid waste as feedstock [30–32].

Catalytic tests were conducted at 700 °C and 800 °C since these temperatures enhanced the generation of non-condensable gas while reducing the tar formation. Calcined dolomite and calcined olivine were used as ex situ catalysts in the pyrolysis cracking catalytic experiments (Figure 2b). It was observed that the cracking effect of calcined olivine led to the production of pyrolytic water, which increased from 11.6 wt.% to 15.9 wt.%, and to a slight reduction in non-condensable gas yield. This can be associated with the enhancement of hydrogenation reactions promoted by this catalyst. On the contrary, the pyrolytic water was decreased due to the high presence of CaO in the dolomite (47.6 wt.%), thus promoting $CO_2$ capture and the water–gas-shift reaction, confirming the dehydration effect associated with this kind of catalyst. Thus, an impressive rise in gas yield was noticed by using calcined dolomite, reaching a maximum of 56.5 wt.% at 800 °C. It was also found that using the regenerated dolomite caused no remarkable differences to be observed in the gas fraction obtained (55.3 wt.%).

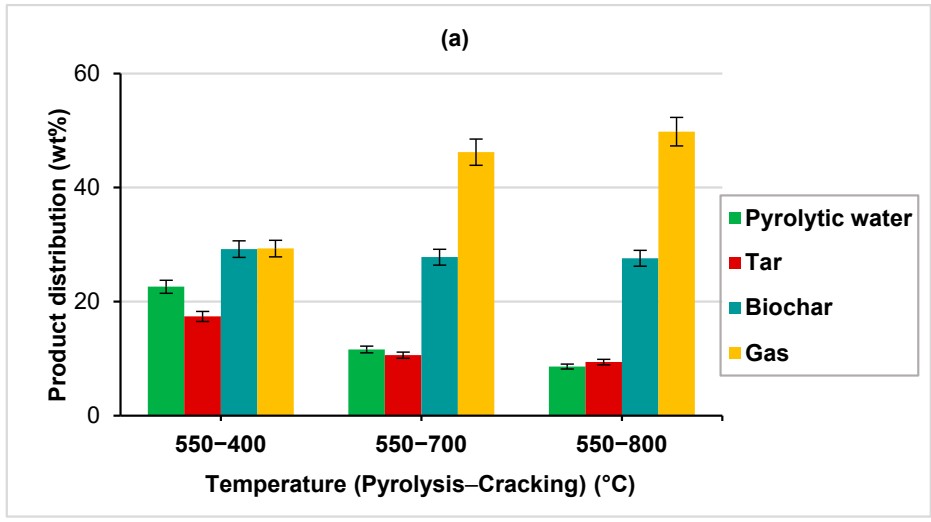

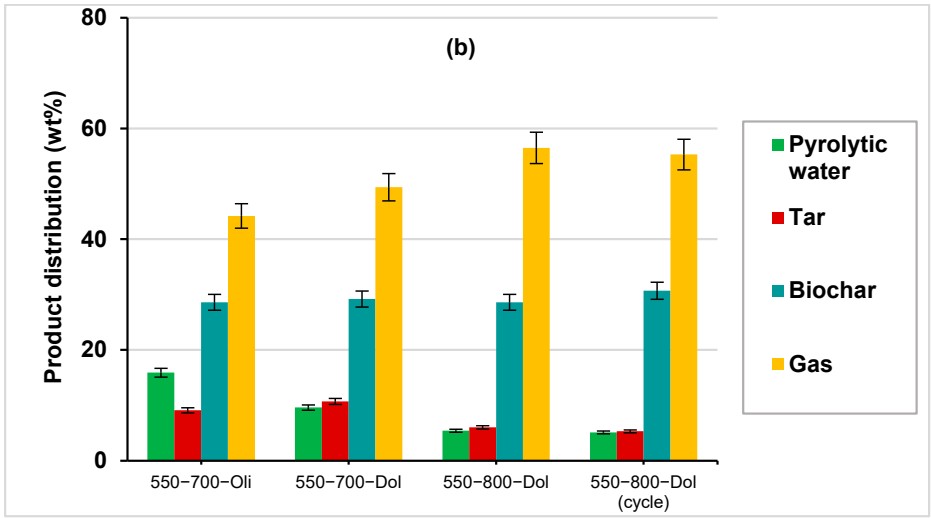

**Figure 2.** (**a**) Effect of pyrolysis and cracking and (**b**) effect of catalytic cracking pyrolysis on different product yields. The values shown in black indicate the closure error of the balance.

Regarding the biochar yields, all the above experiments showed great repeatability, which was expected since the pyrolysis temperature settled at 550 °C in the fixed bed for all the tests. The yield of biochar remained constant within the range of 30 wt.% approximately with slight differences due to the experimental error (RSD ≤ 5%).

2.3.2. Biochar Characterization

The biochar fractions, which were obtained from SGTW pyrolysis at 550 °C, were characterized through the determination of proximate and ultimate analysis, and calorific values, which are summarized in Table 2. The biochar resulting from the SGTW pyrolysis showed a high fixed carbon amount (69.3 wt.%) and lower ash content (8.8 wt.%) compared to that of peat coal. The derived biochar contained high amounts of carbon and nitrogen, which make it a potential fertilizer in different applications that could replenish these organic elements in the soil [33]. The nitrogen content of the biochar surpassed that of the raw SGTW, potentially attributed to the adsorption of different organic compounds containing N- or $N_2$-groups on the char surface [9]. Another noticeable observation is that the hydrogen and oxygen contents on biochar were lower compared to the raw material. This reduction was the result of the dehydration and the loss of carboxyl and hydroxyl (surface functional groups) that produce heavy and light hydrocarbons and non-condensable gas (such as $H_2$, CO, and $CO_2$) during the process of SGTW pyrolysis [3].

These elemental results entailed great calorific values of the derived biochar (26.84 MJ/kg) compared to some coals such as peat coal (22.39 MJ/kg), as well as to the raw material (19.00 MJ/kg).

**Table 2.** Properties of biochar derived from SGTW pyrolysis at 550 °C (peat coal and its biochar are also included for comparative purposes [24]).

|  | SGTW (Biochar) | Peat (Coal) | Peat (Biochar) |
|---|---|---|---|
| **Proximate analysis (wt.%)** | | | |
| Moisture | 6.20 | - | - |
| Ash | 8.80 | 6.51 | 13.54 |
| VM | 5.70 | 69.15 | 19.16 |
| FC * | 79.30 | 24.34 | 67.30 |
| **Ultimate analysis (wt.%)** | | | |
| C | 73.50 | 56.38 | 84.50 |
| H | 2.76 | 5.98 | 2.87 |
| N | 3.81 | 1.43 | 1.18 |
| S | 0.15 | 0.52 | 0.37 |
| O * | 10.98 | 35.69 | 11.08 |
| H/C | 0.45 | 1.27 | 0.41 |
| O/C | 0.11 | 0.47 | 0.10 |
| HHV (MJ/kg) | 26.84 | 22.39 | 31.43 |

* By difference.

To examine the stability and the oxidation degree in the structure of biochar, both the atomic H/C and O/C ratios are plotted in the van Krevelen diagram (Figure 3). It is obviously shown that the ratios of H/C and O/C are lower than 0.6 and 0.4, respectively, indicating that the biochar could be used in the amendment application of soil [9]. Moreover, these atomic ratios of biochar are extremely low compared to the raw material and to peat coal, which is highly desirable to generate energy through combustion (individually or synergistically with fossil fuels) in coal-fired plants. According to Wang et al. [34], the low O/C and H/C ratios of this biochar make its co-firing in powerplants advantageous as it produces less smoke, water vapor, and $CO_2$ when burned, resulting in great combustion efficiency. All these characteristics lead to harnessing the biochar as good-quality fuel in briquette applications, combustion, and even co-combustion with fossil fuels in different energy generation applications, such as running a boiler and generating stationary power.

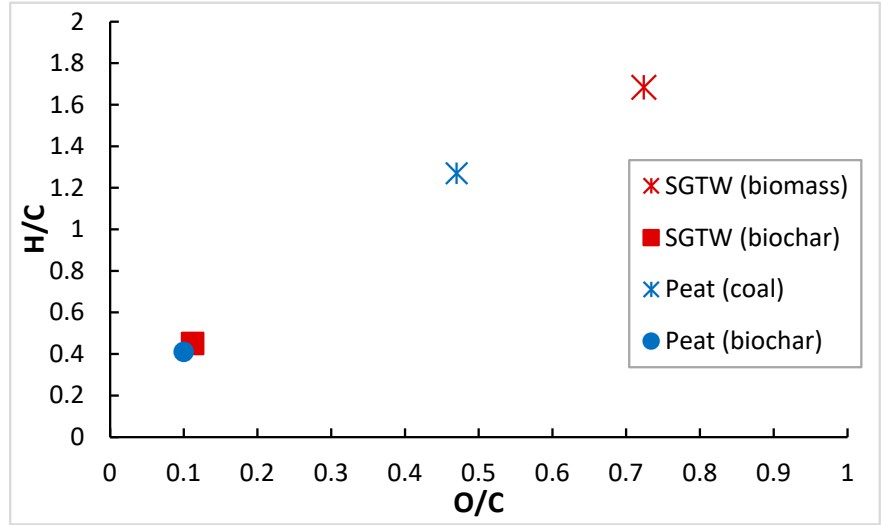

**Figure 3.** Van Krevelen diagram of atomic ratios of SGTW and its derived biochar, peat, and derived biochar at 550 °C.

### 2.3.3. Composition of Non-Condensable Gas

The composition of non-condensable gas was analyzed and is depicted in Figures 4 and 5 (in a free-$N_2$ basis) for different experiments. The resulting gas product consisted of permanent gases ($CO$, $CO_2$, $H_2$, $CH_4$) and light and heavy hydrocarbons ($C_2H_4$, $C_2H_6$, and $C_3$–$C_4$ such as $C_3H_6$, $C_3H_8$, $C_4H_{10}$, $C_4H_8$, and $C_4H_6$).

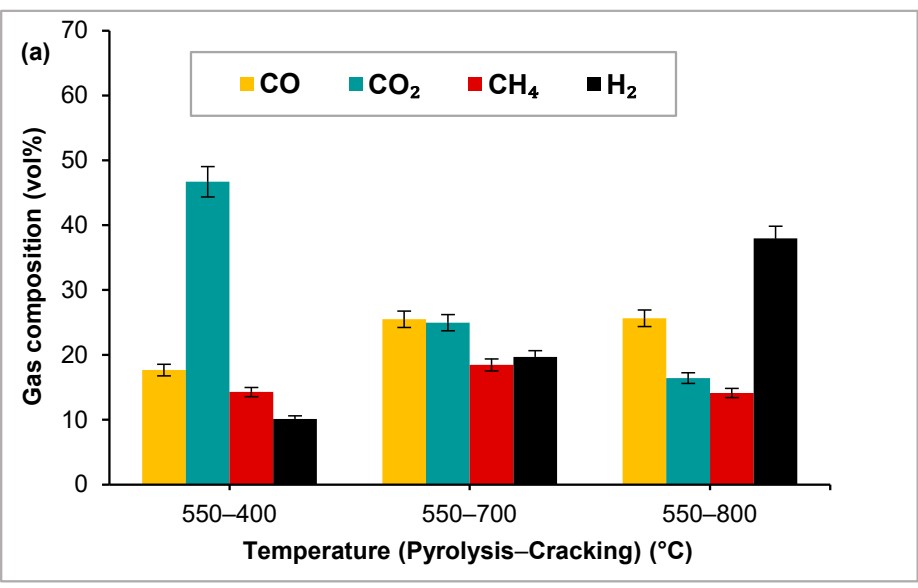

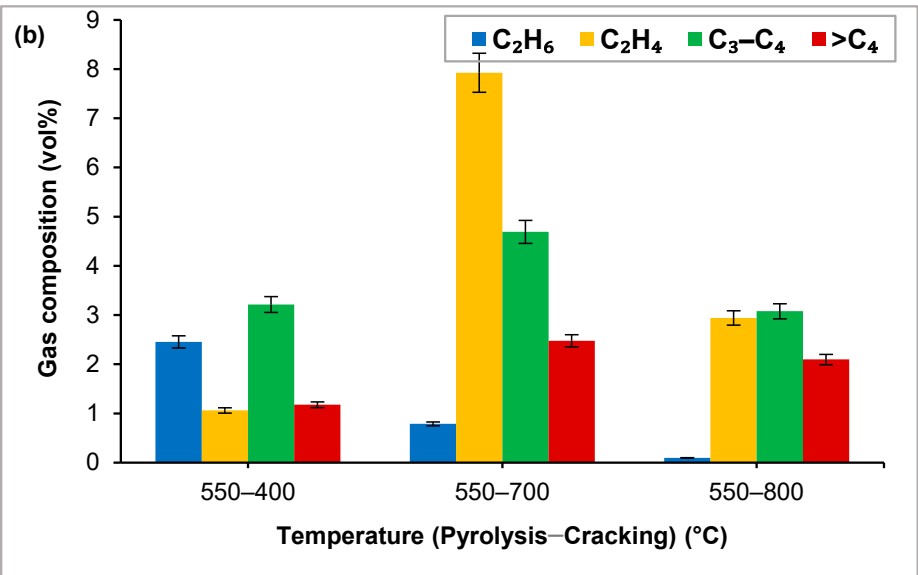

**Figure 4.** Effect of pyrolysis + cracking temperatures on gas composition: (**a**) permanent gases and (**b**) light and heavy hydrocarbons.

During the SGTW pyrolysis at a lower cracking temperature (400 °C), that would correspond with a conventional one-step pyrolysis process, it is noteworthy to emphasize that the $CO_2$ was the primary gas generated followed by $CO$, while $CH_4$ and $H_2$ were also present but in lower concentration. Light hydrocarbons ($C_2H_4$ and $C_2H_6$) and heavier hydrocarbons ($C_3$–$C_4$ and $>C_4$) were also identified in the resulting gas; however, their concentrations were lower (<4 vol%). Focusing on synthetic fuel generation, the M-module (i.e., $H_2$/$CO$ molar ratio) is a significant parameter that studies the suitability of the non-condensable gas in different synthesis processes. Since the $H_2$ content was not significant, the value of $H_2$/$CO$ molar ratio (Table 3) was below 1.0, resulting in the poor potential of this syngas to be used for fuel production.

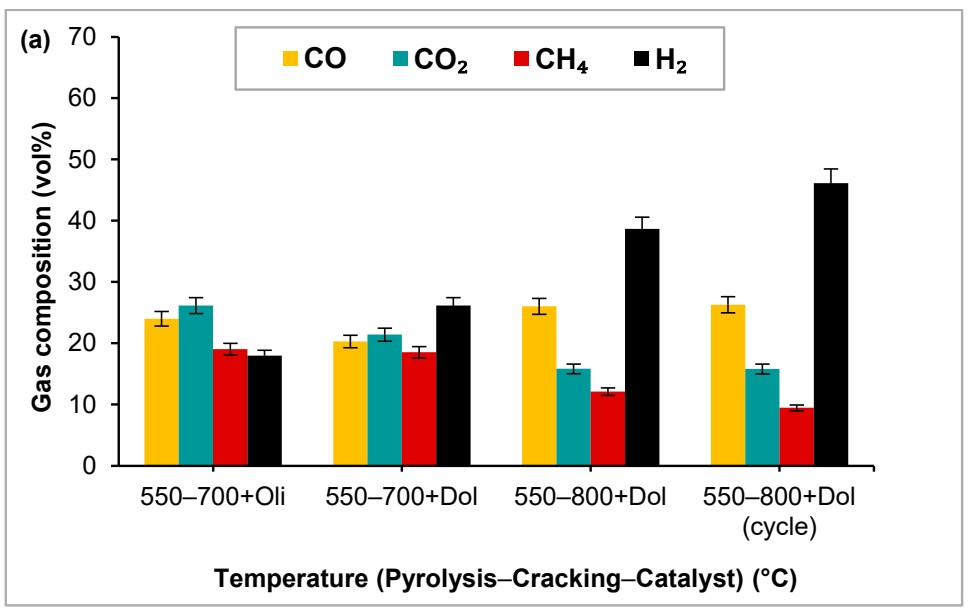

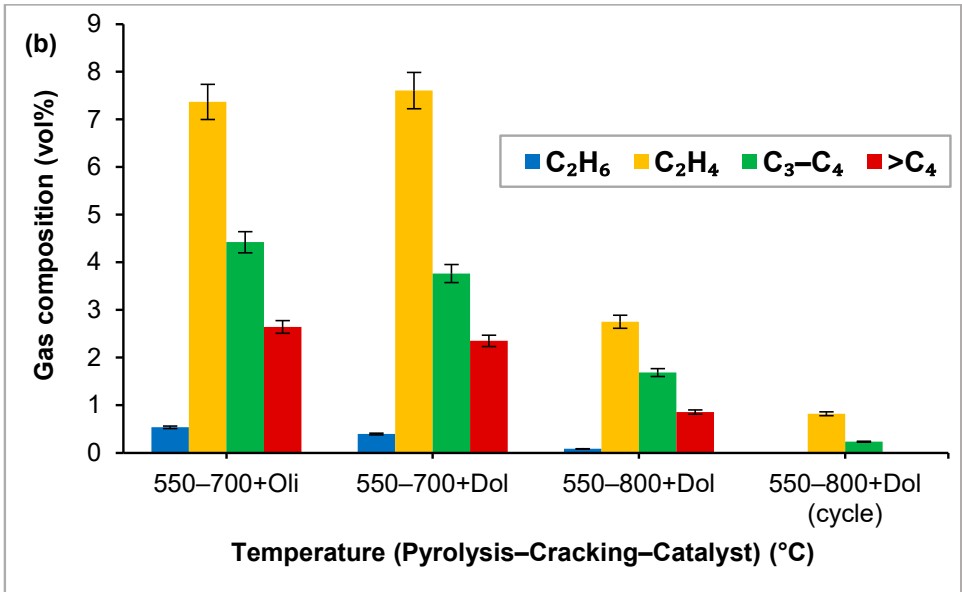

**Figure 5.** Effect of temperatures of catalytic cracking pyrolysis on gas composition: (**a**) permanent gases and (**b**) light and heavy hydrocarbons.

**Table 3.** Properties of syngas from pyrolysis (550 °C) using different cracking temperatures and catalysts.

| $T_{Cracking}$—Catalyst | $H_2$/CO Molar Ratio | CCE (%) | HHV (MJ/Nm$^3$) | ERR (%) |
|---|---|---|---|---|
| 400 °C | 0.57 | 35.30 | 15.2 | 17.72 |
| 700 °C | 0.77 | 61.94 | 21.4 | 47.40 |
| 800 °C | 1.5 | 48.47 | 16.8 | 52.99 |
| 700 °C—Olivine | 0.75 | 56.59 | 19.8 | 40.43 |
| 700 °C—Dolomite | 1.3 | 63.91 | 19.9 | 59.15 |
| 800 °C—Dolomite | 1.5 | 46.97 | 15.9 | 59.68 |
| 800 °C—Dolomite-Cycle | 1.8 | 33.00 | 13.8 | 49.65 |

Raising the temperature inside the cracking reactor from 400 °C to 700 °C and 800 °C resulted in a significant decrease in the concentration of $CO_2$ (from 46.71 vol% down to 24.96 vol% and 16.34 vol%, respectively). In contrast, the $CH_4$ composition was increased

followed by a slight reduction at 800 °C. The presence of $H_2$ was also enhanced, reaching values more than three-fold, 37.96 vol%, in comparison to the value obtained at low cracking temperature, due to the secondary reactions of bio-oil. Finally, the concentration of CO sharply increased from 17.66 vol% up to 25.65 vol%. These outcomes suggested that a significant proportion of vapors containing different high-molecular-weight compounds were mostly cracked into light components $CH_4$, CO, and $H_2$, prompting a substantial rise in the non-condensable gas yield (see Figure 5). The concentrations of $C_3$–$C_4$, >$C_4$, and $C_2H_4$ fluctuated by increasing the cracking temperature, but the $C_2H_6$ percentage decreased from 2.46 vol% to 0.10 vol %. The maximum of $C_3$–$C_4$, >$C_4$, and $C_2H_4$ yields was reached at 700 °C, where $C_2H_4$ featured a noticeable increase (approximately seven times higher). Due to the dramatic increase in ethylene content, syngas becomes more valuable, since $C_2H_4$ is an important product in the chemical and petroleum industries. Although the rise in cracking temperature boosted the formation of $C_2H_4$ and $C_2H_6$, the highest temperature led to further active Diels–Alder reactions and dehydrogenation reactions, resulting in the production of more $H_2$ from $C_2H_6$ and $C_2H_4$. Consequently, the concentration of $C_2H_6$ and $C_2H_4$ decreased when the temperature was raised beyond 700 °C. The distribution of final components in the gas fraction resulting after the two thermal cracking experiments suggests that the main reactions taking place were the water–gas-shift (WGS), tar reforming, methane steam reforming, and Boudouard reactions. These different reactions, promoted especially at 800 °C [32,35], are as follows:

$$\text{Water gas-shift: } CO\,(g) + H_2O\,(g) \rightleftharpoons CO_2\,(g) + H_2\,(g) \qquad\qquad \Delta H_{298K} = -41\text{ kJ/mol} \qquad (1)$$

$$\text{Tar reforming: } Tar \rightarrow CO\,(g) + CO_2\,(g) + H_2\,(g) + C_nH_m\,(g) \qquad\qquad\qquad\qquad (2)$$

$$\text{Methane steam reforming: } CH_4\,(g) + H_2O\,(g) \rightleftharpoons CO\,(g) + 3\,H_2\,(g) \qquad \Delta H_{298K} = +206\text{ kJ/mol} \qquad (3)$$

$$\text{Boudouard reaction: } C\,(s) + CO_2\,(g) \rightleftharpoons 2\,CO\,(g) \qquad\qquad \Delta H_{298K} = +172\text{ kJ/mol} \qquad (4)$$

The non-condensable gas released at higher cracking temperatures, particularly at 800 °C, can be considered a potential source of green $H_2$ for some industrial applications. The significant amount of $H_2$ in the gas stream led to higher $H_2$/CO molar ratios, which achieved 1.5 at 800 °C (Table 3). This obtained gas is of wide interest to chemical industries. As reported by Suttikul et al. [36], various chemical products, such as isobutanol, isobutene, higher alcohols ($C_1$–$C_6$), and aldehydes, can be generated from syngas characterized by an $H_2$/CO ratio of 1.5.

Various tendencies were pointed out regarding the use of dolomite and olivine as ex situ catalysts during the cracking pyrolysis of SGTW. According to Figure 5a, the catalytic tests at 700 °C obviously affected the distribution of the non-condensable gas compounds. The use of calcined olivine at 700 °C enhanced the production of $CO_2$ (26.16 vol%), while $H_2$ and CO suffered a slight reduction from 19.67 vol% to 17.98 vol% and from 25.5 vol% to 24 vol%, respectively. In contrast, using dolomite showed an opposite influence on the composition of the gas stream. At 700 °C, both of the concentrations of $CO_2$ and CO reduced, whilst that of $H_2$ rose and that of $CH_4$ was not affected. Raising the cracking temperature when using calcined dolomite sharply increased the amount of the CO and $H_2$ gas stream and reduced the $CO_2$ and $CH_4$ concentrations. Consequently, the important catalytic impact of magnesium and calcium species in the calcined dolomite was confirmed through the enhancement of the $CO_2$ absorption to further raise the concentration of $H_2$, leading to a great $H_2$/CO molar ratio (1.3 and 1.5 at 700 °C and 800 °C, respectively). On the other hand, a lower $H_2$/CO ratio (below 1.0) was observed while using the calcined olivine in the cracking pyrolysis tests, resulting in a poor gas with its worst potential application. The presence of dolomite during the cracking experiments promoted different reactions,

such as WGS, dry reforming, and carbonization [32], which preferentially occurred in the presence of catalysts, reducing the concentrations of $C_3$–$C_4$, >$C_4$, $C_2H_4$, and $C_2H_6$.

$$\text{Dry reforming reaction: } C_nH_m + nCO_2 \rightleftharpoons 2nCO + (m/2)\,H_2 \qquad \Delta H_{298K} > 0 \qquad (5)$$

$$\text{Carbonization reaction: } C_nH_{2n+2} \rightarrow nC + (n+1)\,H_2 \qquad \Delta H_{298K} > 0 \qquad (6)$$

The evolution of syngas composition was also assessed after the use of regenerated dolomite. It should be noted that, although minor concentration values of $C_2$, $C_3$, and >$C_4$ compounds were identified (discrepancies that could be attributed to experimental variability), these species were reduced after these experiments. Observations from Figure 5a revealed the preservation of a minimal quantity of $CO_2$ in the gas stream, alongside a minor elevation in the CO levels. Simultaneously, the abundance of $H_2$ (46.14 vol%) remained substantial, underscoring the sustained catalyst activity. As a result, over half of the total gas stream comprised $H_2$ and CO (72.4 vol%).

These values translated into a $H_2$/CO molar ratio within the same range of the fresh dolomite, even reaching the maximum value (1.8). Therefore, this obtained gas retained its potential to be used as feedstock for the generation of various fuels and chemicals.

From Table 3, it was observed that as the cracking temperature increased from 400 °C to 700 °C, the carbon conversion efficiencies (CCEs) sharply rose due to the significant conversion of SGTW to non-condensable gas through cracking water–gas-shift and Boudouard reactions. As CCE represents the fraction of carbon content in the biomass that was converted to different gas products such as $CO_2$, CO, $CH_4$, and $C_2H_4$, its values changed by increasing the cracking temperatures and using different calcined catalysts. The maximum value of CCE (63.91%) was reached at 700 °C through the use of calcined dolomite as an ex situ catalyst. This result was expected owing to the great concentration of $C_2H_4$ and heavier hydrocarbons under these conditions, in addition to the high gas yield.

The HHV of gases generated during the cracking pyrolysis increased, raising the temperature and reaching the maximum of 21.4 MJ/Nm$^3$ at 700 °C, followed by a reduction (16.8 MJ/Nm$^3$) at 800 °C (Table 3). These results were ascribed to the variation in the gas compositions with the increment in cracking temperatures. In fact, $CH_4$, $C_2H_6$, and heavy hydrocarbon contents in the gases increased with the rise in temperature from 400 to 700 °C in the cracking reactor (see Figure 4). However, at a higher cracking temperature (800 °C), the concentrations of the three main components reduced, ultimately decreasing the higher heating value of non-condensable gas. The use of catalysts in the cracking reactor led to a reduction in the HHV of gases, which was expected as light and heavy hydrocarbons sharply decreased (see Figure 5). This was due to the fact that more $H_2$ was generated than in the non-catalytic tests, whose volumetric energy density was small. The values of HHV of gases obtained after catalytic and non-catalytic cracking were significantly higher, compared to those resulting from SGTW pyrolysis, hence raising the fuel gas potential to be further involved in the power generation applications. Similar tendencies have been shown in the literature [31].

The amount of energy recovered (ERR) from the feedstock is also a significant parameter in assessing the potential quality of gas products and the process viability. It is calculated through Equation 13 and summarized in Table 3. Regarding the non-catalytic cracking pyrolysis, it is mentioned that the energy recovery rate of the gas product sharply increased with the rise in cracking temperature from 400 to 800 °C. In addition, using catalysts during the cracking pyrolysis also affected the value of ERR. As appreciated in Table 3, the great values of ERR corresponded to the cracking process executed using calcined dolomite as a catalyst (59.15% and 59.68% of those introduced with SGTW at 700 °C and 800 °C, respectively). On the other hand, it should be remarked that the ERR value after the use of regenerated dolomite at 800 °C was not highly affected. Consequently, although the thermal process's energy requirements must be considered, it is worth emphasizing that up

to 59% of the energy recovery value were resulted when calcined dolomite was introduced as a cracking catalyst at 800 °C, indicating that most of the energy released from the reaction of catalytic cracking pyrolysis of SGTW was retained in the syngas. Hence, it is probed that catalytic cracking pyrolysis could be a great solution to improve the quality of gas products to be used as syngas in further industrial applications.

### 2.4. Catalyst (Dolomite) Characterization

The purity and crystallinity of dolomite, as the best low-cost catalyst to improve the gas product quantity and quality, was verified through X-ray diffraction. Dolomite primarily consists of $CaCO_3$ and $MgCO_3$ with minor traces of Fe, Si, and Al, as well as different traces of various mineral impurities [37]. Three different XRD patterns of dolomite (fresh, used after cracking pyrolysis at 800 °C, and regenerated (1 cycle)) are depicted in Figure 6. In fresh dolomite, the diffraction peaks of Ca $(OH)_2$ (2Theta = 34), possibly originating from absorbed water during handling, and CaMg $(CO_3)_2$ (2Theta = 47 and 50), were evidenced. However, these peaks vanished when dolomite was used during the cracking process and subsequently regenerated. Consequently, the CI decreased from 68.5% to 58.8% (as shown in Table 4). The decomposition of CaMg $(CO_3)_2$ and Ca $(OH)_2$ led to the generation of CaO and MgO, which constituted the major components in the used and regenerated dolomite. However, the diffraction patterns of these components exhibited few diffraction peaks similar to those present in fresh dolomite, indicating the incorporation of additional CaO. This result correlates with the significant rise in $H_2$ and reduction in $CO_2$, as the $CO_2$ capture linked to CaO promotes the production of $H_2$ through the water–gas-shift (WGS) reaction [38]. Therefore, although additional cycles are required to comprehensively evaluate the cyclic catalytic stability of the catalyst, these results are optimistic since the application of this kind of regenerated catalyst is not negatively affected from an economical and energetical point of view to the process [39,40].

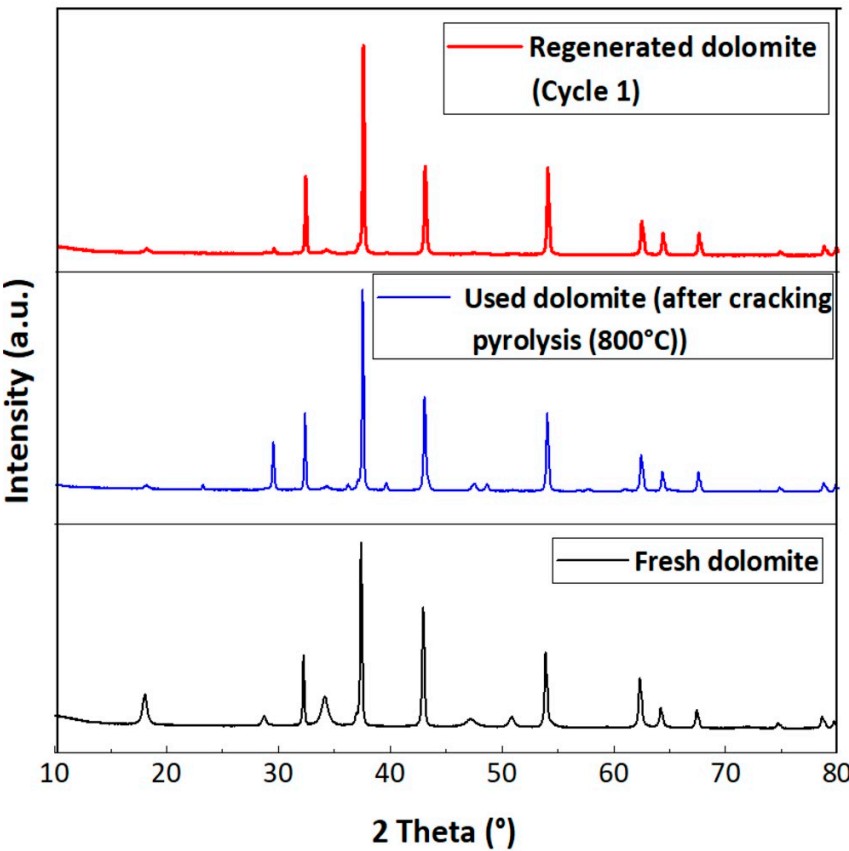

**Figure 6.** XRD patterns of fresh and used dolomite.

**Table 4.** Crystallinity index of fresh, used, and regenerated dolomite.

| Catalyst<br><br>Crystallinity Index | Fresh Dolomite | Used Dolomite (after Cracking Pyrolysis (800 °C)) | Regenerated Dolomite (Cycle 1) |
|---|---|---|---|
| CI (%) | 68.5 | 63.6 | 58.8 |

### 2.5. Reliability of Catalytic Cracking Pyrolysis Process of Biomass

The catalytic cracking pyrolysis of different types of coffee waste (pure spent coffee ground (PSCG) and spent coffee ground blended with 50% of torrefied barley (BSCG)) was also studied to investigate the efficiency and reliability of the process and its influence on the quantity and quality of gas products. Despite the different physicochemical properties of tea and coffee wastes (SGTW, PSCG, and BSCG), similar results were observed during catalytic and non-catalytic cracking pyrolysis at 550 °C. From Table 5, it is obvious that the gas yields increased by more than half with the rise in cracking temperatures and also continued increasing after adding calcined and regenerated dolomite at 800 °C.

**Table 5.** Properties of gas products derived from different coffee waste biomass under different experimental conditions using a pyrolysis temperature of 550 °C.

| | $T_{Cracking}$—Catalyst | Gas Yield (wt. %) | $H_2/CO$ | CCE (%) | HHV (MJ/Nm$^3$) | ERR (%) |
|---|---|---|---|---|---|---|
| **PSCG** | 400 °C | 26.9 | 0.7 | 31.8 | 20.7 | 20.2 |
| | 800 °C | 50.5 | 1.6 | 50.7 | 19.7 | 48.6 |
| | 800 °C—Dolomite | 60.9 | 1.5 | 47.6 | 18.3 | 63.0 |
| | 800 °C—Dolomite-Cycle | 67.0 | 1.6 | 39.5 | 14.4 | 63.8 |
| **BSCG** | 400 °C | 25.9 | 0.5 | 28.5 | 19.6 | 17.6 |
| | 800 °C | 53.8 | 1.6 | 47.4 | 17.8 | 43.7 |
| | 800 °C—Dolomite | 57.8 | 1.6 | 41.2 | 17.9 | 58.6 |
| | 800 °C—Dolomite-Cycle | 61.9 | 1.7 | 32.3 | 13.7 | 57.1 |

Focusing on the composition of different gas products (Figure 7), equivalent results were observed at the same experimental conditions for the different types of biomass waste. The concentration of $CO_2$ suffered drastic reductions. Therefore, the resulting gases with an HHV that ranges from 14 to 21 MJ/Nm$^3$ could be recycled and burnt to operate pyrolysis systems and could also be used as a fuel in the combustion processes addressing notable environmental issues. Moreover, the different gas obtained during catalytic cracking pyrolysis, especially while using dolomite, can be considered potential sources of green hydrogen for some industrial applications. As a potential intermediate application, these products can be used in engines or boilers to produce electricity in small facilities. The great amount of $H_2$ (around 50 vol%) led to highly interesting $H_2/CO$ ratios, ranging from 1.5 to 1.7, which are often used in the production of syngas-based valuable chemicals [41] and transportation fuels via Fischer–Tropsch synthesis [42]. Compared to CCE obtained during catalytic cracking pyrolysis of SGTW, similar results were shown for those of PSCG and BSCG owing to the decline in concentrations of light and heavy hydrocarbons. Moreover, it must be highlighted that 63.8% and 58.6% of the energy generated from the catalytic (dolomite) cracking pyrolysis at 800 °C of PSCG and BSCG, respectively, were kept in the gas product.

Finally, after applying the catalytic cracking pyrolysis on different biomass wastes, it is worth stating that this process could be considered a simple and clean solution to valorize lignocellulosic biomass and generate valuable gaseous by-products.

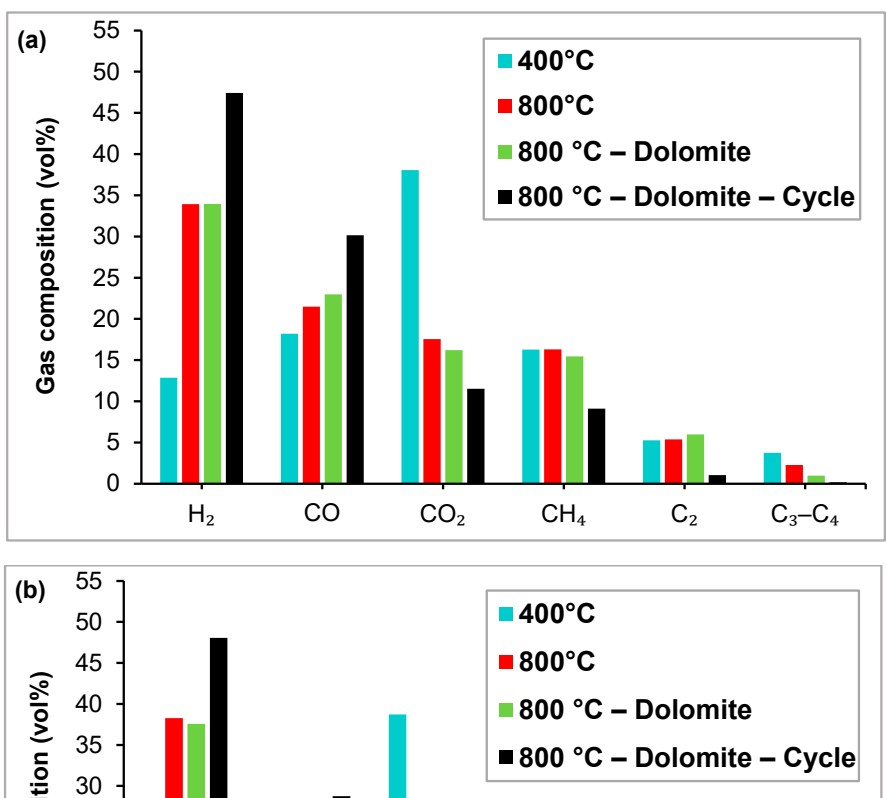

**Figure 7.** Impact of cracking temperatures and catalyst (calcined and regenerated dolomite) at 550 °C on gas composition of (**a**) PSCG and (**b**) BSCG.

## 3. Material and Methods

### 3.1. Biomass Waste Characterization

The biomass waste employed in the current study was SGTW collected from various cafeterias in the Tunis region after soaking the green tea in hot water for beverage preparation. The SGTW was naturally air-dried for 24 h and then stored for different analyses and pyrolysis experiments. The proximate analysis of the feedstock was performed using different analytical instruments: UNE-EN ISO 18134 for moisture, UNE-EN ISO 18122 for ash content, and UNE-EN ISO 18123 for volatile matter (VM); the fixed carbon (FC) was determined through balance. The inorganic elements presented in the ash were quantified using the multi-element analysis technique, which uses an inductive coupled plasma source (ICP-OES). Ultimate analysis (CHNS) of the sample was carried out using a Thermo flash 1112 (according to UNE EN 5104), while the oxygen content was calculated through difference. In order to study the bioenergy potential of biomass, it is significant to calculate its calorific value. The higher heating value HHV (MJ/kg) was calculated based on Channiwala and Parikh's correlation [43] using Equation (7) below:

$$HHV = 0.3491\,(C) + 1.1783\,(H) + 0.01005\,(S) - 0.1034\,(O) - 0.0151\,(N) - 0.0211\,(Ash) \tag{7}$$

Two types of coffee wastes, pure spent coffee ground (PSCG) and blended spent coffee ground with 50 wt% of torrefied barley (BSCG), were used in this current study for the verification of the effectiveness and reliability of the catalytic cracking pyrolysis process. They were collected from different coffee shops [28].

### 3.2. Catalysts

Two low-cost commercial catalysts were used in the current study: dolomite ($MgO.CaO$) and olivine ($MgO.SiO_2.Fe_2O_3$). The primary constituents of the dolomite were calcium oxide (CaO, 47.6 wt%) and magnesium oxide (MgO, 33.2 wt%). In contrast, the other observed elements were present in amounts less than 1.0 wt% (including $Al_2O_3$, $K_2O$, and $SiO_2$). Conversely, olivine exhibited a significant content of magnesium oxide (MgO, 51.6 wt%) and a substantial content of silicon dioxide ($SiO_2$, 36.5 wt%), along with a notable concentration of iron (III) oxide ($Fe_2O_3$, 10.2 wt%). The remaining components, such as CaO, copper oxide (CuO), nickel oxide (NiO), manganese oxide (MnO), or chromium (III) oxide ($Cr_2O_3$), were found in lower concentrations, all below 1 wt%.

These natural mineral catalysts were calcined in static air furnace at 875 °C using a high heating rate (30 °C/min) for 2 h. A study of the lifetime activity of the dolomite was also conducted. For this reason, a test was conducted including the SGTW catalytic cracking pyrolysis at 800 °C, followed by the regeneration of dolomite in a static air furnace (875 °C, 30 °C/min, 2 h). Once dolomite regeneration was completed, it was recovered and reincorporated into the cracking pyrolysis facility, accomplishing one entire cycle of dolomite.

### 3.3. Thermogravimetric Analysis

Thermogravimetric analysis of SGTW was performed in a Netzsch Libra F1 Thermobalance in order to examine the thermal behavior of SGTW under different pyrolysis conditions. The raw material (10 mg) was heated from room temperature to 800 °C using a heating rate of 25 °C/min under a $N_2$ atmosphere (50 NmL/min) to avoid the unwanted oxidation of the sample. The temperature and the weight loss changes of solid were recorded. The derivative thermogravimetry (DTG) curve of SGTW was deconvoluted using the Gaussian-type signal to determine the amounts of all constituents (extractives, hemicelluloses, cellulose, and lignin) [29].

### 3.4. Two-Stage Process

Experiments of SGTW pyrolysis and cracking were carried out in a lab-scale two-stage facility involving a vertical fixed bed reactor composed of stainless steel (52.5 cm height and 5 cm internal Ø) linked with a horizontal tubular reactor (29.5 cm length and 1.5 cm internal Ø) (Figure 8 [30,44]). The fixed bed was featured by a vertical mobile liner, where 16 g of spent green tea was introduced. Therefore, it was feasible to preheat the reactor to the required temperatures, while contact with the samples was avoided. Once the desired temperature was reached in the reactor, the mobile liner was propelled to the reaction zone, assuring the rapid heating rates needed for the process of devolatilization (~100 °C/min). The raw material was pyrolyzed using nitrogen (400 mL/min) to remove the volatile vapors continuously from the system and to preserve an inert atmosphere inside the reactor. In order to guarantee the total devolatilization of SGTW, 25 min was selected as reaction time. The non-condensable and condensable gases passed through a tailor-made condenser with the use of water reflux at 7 °C to collect the maximum liquid product after reaction. The tar fraction, which presented the organic liquid fraction, was mainly located in different parts of the system (cracking reactor, condenser, and filter). Therefore, the yield of tar was determined by measuring the difference in weight of the installation parts before and after each experiment. The non-condensable gas was collected in a Tedlar sampling bag situated after a filter to purify the gas mixture for further analysis. At the end of every experiment, the biochar (solid fraction) was taken out from the liner

after reaching the ambient temperature inside the reactor. The yields of pyrolysis products (solid, gas, and liquid) were procured by weight:

$$\text{Solid yield (\%)} = (\text{mass of biochar}/\text{mass of SGTW}) \times 100 \tag{8}$$

$$\text{Gas yield (\%)} = (\text{mass of gas}/\text{mass of SGTW}) \times 100 \tag{9}$$

$$\text{Liquid yield (\%)} = (\text{mass of liquid}/\text{mass of SGTW}) \times 100 \tag{10}$$

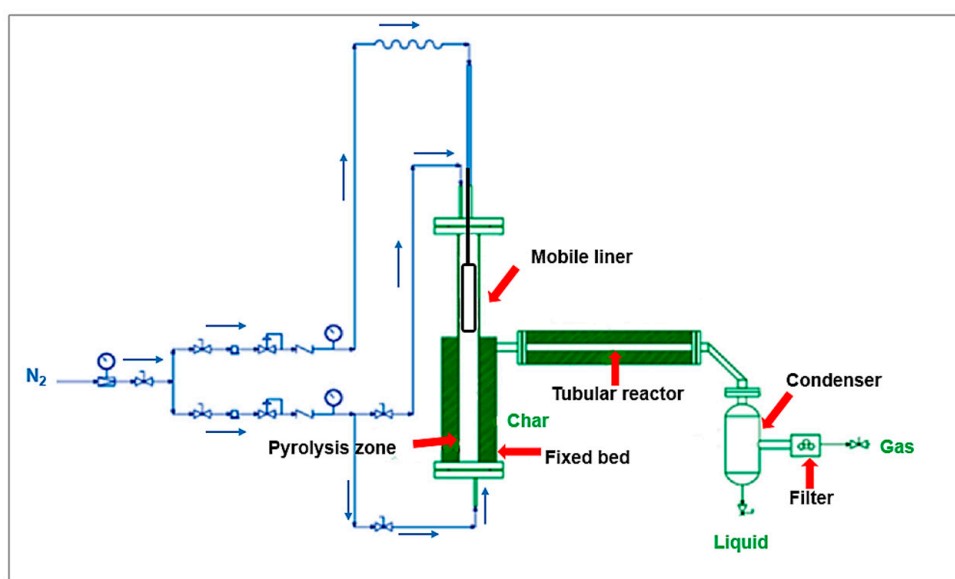

**Figure 8.** Schematic diagram of a two-stage pyrolysis reactor.

During all the cracking pyrolysis experiments, the temperature inside the fixed bed was adjusted to 550 °C, while three temperatures (400, 700, and 800 °C) were settled in the cracking reactor to assess the effect of the thermal cracking of gas products from SGTW pyrolysis. Moreover, the performance of two catalysts (olivine and dolomite) was studied in the cracking pyrolysis process, keeping steady the sample/calcined catalyst ratio of 1:1. A catalytic pyrolysis cycle of dolomite was carried out to evaluate the impact of dolomite regeneration on the quantity and quality of different products. Experiments with two types of coffee waste were also performed to evaluate the application of the process. To ensure the reproducibility of results, pyrolysis tests were performed twice and the average values are reported.

*3.5. Characterization of Process Products and Catalyst (Dolomite)*

The solid and gaseous products obtained from the pyrolysis of SGTW were analyzed using different analytical techniques.

The biochar was characterized by determining its proximate and ultimate analysis according to the analytical standards mentioned in Section 3.1, and by calculating its calorific value (Equation (7)).

The non-condensable gaseous fraction generated was analyzed using two different gas chromatographs (GC). The permanent gases (such as $H_2$, $O_2$, $N_2$, and CO) were analyzed in a Bruker 450-GC equipped with a TCD detector. The chromatograph was outfitted with two SS packed columns (a Molsieve 13X and a HayeSep Q). The oven program used was 60 °C during 10 min. The TCD and injector temperatures were 200 °C. Light hydrocarbons (methane, ethane, isobutene, etc.) were determined through a Hewlett Packard 5890 series II GC equipped with an Alumina Chloride PLOT capillary column (30 m, 0.32 mm) and an FID detector. The programmed temperature method used during this type of analysis was

isothermal at 50 °C for 7 min and then an implemented heating rate of 25 °C/min to reach the final oven temperature of 140 °C that was maintained for 5 min. The FID and injector temperatures were 220 °C and 150 °C, respectively. Thus, gas chromatographs (GCs) assume the role of a paramount tool for separating volatile or vaporizable compounds that can be quantified. Furthermore, it serves as an indispensable means for quantifying various components within a complex mixture.

The crystal phases present on the fresh dolomite and the obtained dolomite, after cracking pyrolysis at 800 °C and after 1 cycle, were analyzed using X-ray diffraction (Bruker D8 Advance series II diffractometer, Cu-Ka radiation (k = 0.1541 nm). The patterns of X-ray diffraction (XRD) were recorded over a 2 Theta (2θ) angle range of 10–80° using a scan speed of 1°/min. XRD is a versatile technique for elucidating the crystal structure of a catalyst, encompassing both the fresh and regenerated dolomite samples. This technique provides invaluable insights into the crystallinity and structure of the catalyst, information that remains elusive through other analytical methods.

*3.6. Data Analysis*

Carbon conversion efficiency (CCE) represents the fraction of carbon in raw material (SGTW), which was converted to various gas products (such as $CO_2$, $CO$, $CH_4$, and $C_nH_m$) [45]. The CCE was determined as follows:

$$CCE\ (\%) = [12Y \times (\%C\text{-}CO + \%C\text{-}CO_2 + \%C\text{-}CH_4 + n \times C\%\text{-}C_nH_m) \times 100]/(22.4 \times \%C\text{-}biomass) \tag{11}$$

where Y is the yield of gas product ($m^3$/kg); $\%C\text{-}CO$, $\%C\text{-}CO_2$, $\%C\text{-}CH_4$, and $\%C\text{-}C_nH_m$ represent the molar carbon (%) of different components in the gas product ($CO$, $CO_2$, $CH_4$ and $C_nH_m$ (light and heavier hydrocarbons), respectively); and %C-biomass is the mass of carbon (%) obtained from the ultimate analysis of the raw material.

The higher heating value (HHV) of the non-condensable gas was evaluated using Equation (12) [46]:

$$HHV_g\ (MJ/Nm^3) = C_1 \times HHV_1 + C_2 \times HHV_2 + \ldots + C_n \times HHV_n \tag{12}$$

where $C_1$, $C_2$, ... $C_n$ represent the volume fractions of different gas components, and $HHV_1$, $HHV_2$, ... ,$HHV_n$ are the related high heating value of the same gas components.

The energy recovery ratio (ERR), which is defined as the percentage of non-condensable gas gross energy [47], is expressed as follows (Equation (13)):

$$ERR\ (\%) = [(yield\ of\ syngas \times HHV_{syngas}) \times 100]/HHV_{biomass} \tag{13}$$

The crystalline index (CI), which was determined from the XRD data, is expressed using the formula [48]:

$$CI\ (\%) = A_c \times 100/\ (A_c + A_a) \tag{14}$$

where $A_c$ and $A_a$ represent the area under the different crystalline peaks and the area of the amorphous hollows, respectively.

## 4. Conclusions

A novel two-stage process was conducted to manage SGTW by combining pyrolysis with thermal catalytic and non-catalytic cracking under a non-oxidizing atmosphere in a lab-scale system. The positive impact of incorporating cost-effective catalysts such as olivine and dolomite as ex situ catalysts during the cracking step should be highlighted. Especially, using calcined dolomite at high pyrolysis cracking temperatures led to a reduction in poor-quality bio-oil production with a substantial increase in the syngas yield and quality. Thus, a CO and $H_2$-rich gas can be obtained, potential as feedstock for the generation of various fuels and chemicals ($H_2$/CO molar ratio > 1.5). In addition, a high-quality solid fuel with an HHV of 26.84 MJ/kg was also produced, suitable for applications such as briquettes, combustion, and/or co-combustion with fossil fuels in various energy generation processes.

Similar outcomes were observed when applying the catalytic cracking pyrolysis process on different types of coffee waste, revealing that this process is a simple and clean solution to valorize lignocellulosic biomass and generate valuable gaseous products. Although more cycles are needed for the consolidation of this process, an initial approach using regenerated dolomite after the process demonstrated that gas yields and composition can be kept practically unaltered.

**Author Contributions:** A.B.A., investigation, data curation, formal analysis, writing—original draft. A.B.H.T., supervision, writing—review and editing. A.V., data curation, methodology, writing—review and editing, investigation. T.G., supervision, writing—review and editing, conceptualization, funding acquisition, supervision. J.M.L., investigation, writing—review and editing. M.V.N., data curation, writing—review and editing. D.M., supervision, writing—review and editing. All authors have read and agreed to the published version of the manuscript.

**Funding:** This work was supported and funded by the Ministry of Higher Education and Scientific Research (MESRST) in Tunisia (Program 2020–2021). The financial support for this work was also given by the Grant PID2021-123759OB-I00 funded by MCIN/AEI/ 10.13039/501100011033, by "ERDF A way of making Europe", as well as by the Regional Government of Aragon (DGA).

**Data Availability Statement:** The data presented in this study are available on request from the corresponding author.

**Acknowledgments:** The authors acknowledge Ministry of Higher Education and Scientific (MESRST), the Spanish Ministry of Science and Innovation, the State Research Agency and the Regional Government of Aragon (DGA) under the research groups support programme.

**Conflicts of Interest:** The authors declare no conflict of interest.

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
