# Peer review of "Enhancing the Production of Syngas from Spent Green Tea Waste through Dual-Stage Pyrolysis and Catalytic Cracking"

_catalysts, doi:10.3390/catal13101334_

Round 1
Reviewer 1 Report
This manuscript investigated the catalytic cracking of SCTW for producing gas product. The effect of catalysts addition and catalytic cracking temperature was investigated in detail. The obtained bio-char was analyzed. The feasibility of this catalytic cracking process for other biomass was also studied. In general, this manuscript can be accepted for publication after considering the following issues:
1. Please provide the detailed process of catalytic pyrolysis cycle.
2. The formula for calculating the yield of each products should be provided.
3. In Figure, y-axis “Mass loss” can be revised as “Mass”.
4. How to separate the pyrolytic water and organic fraction?
5. In Figure 3a, what’s the meaning of 1.5/3.8/4.6?
6. The experimental error for each products can be provided.
7. The reason for the decrease of pyrolytic water should be discussed.
8. More cycle performance of Dol catalyst should be provided.
The English is well.
Author Response
Please, find attached the detailed responses

Reviewer 2 Report
In this work, Abdallah et. al have investigated production of syngas from spent green tea waster via pyrolysis and subsequent catalytic upgrading process over two low-cost commercial catalysts, namely Dolomite and Olivine. Their results demonstrate that the calcined dolomite catalyst reduced the bio-oil yield, meanwhile enhanced the syngas yield and the maximum potential energy of the stream. The manuscript is well written. This work is of interest to the audience of catalysts in general. I have only few comments that needs to be addressed before acceptance.
1. The results demonstrate that Dolomite performances better than Olivine. However, no explanation regarding their difference is offered in the manuscript. Can the authors comment on the chemical components of these two catalyst, MgO.CaO vs MgO.SiO2.Fe2O3? What is the content of non-catalytic SiO2 in Olivine? Will that affect the performance of the catalyst if the same amount of catalyst is loaded?
2. There have been lots of publications in the literatures on biomass pyrolysis using metal oxides such as Fe2O3 and CaO as the catalysts, which might provide some insights into the difference between Olivine and Dolomite.
Author Response

(The authors gave the same response as above.)

Reviewer 3 Report
The article handles about the use of catalysts for the production of fuels.
(1) I don't think that seaweed as a marine biomass has a very low content on nitrogen and sulphur. "Biomass has been considered as a potential energy source, which can be used to generate various materials and chemicals. Compared to conventional fossil fuels, biomass is a clean CO2 neutral alternative, and it also contains low amounts of nitrogen and sulfur [1]." It is very important to provide a comprehensive overview over the previous field. Try to avoid generalisation and rather start with the paragraph that shows what why it is necessary to use fuels in general. Currently, we have electro-cars and also other renewable sources. Also cost of biofuel should be taken into the discussion and potential to use biofuel possibly in marine engines as the most viable option compared to busses, cars, etc.
(2) Instead of explaining what thermochemical methods mean, it would be good to summarize the previous research on catalysts and why the catalysts are necessary. Also prior you will underline the novelty of your work, please, add a sentence "The gap of the literature is based on ..." Show what has been known in the literature and what remained unknown.
(3) Emphasize the importance of catalyst use and emphasize the novelty of you.
(4) "... processes).7 A ... " or "... of the ctalytic cracking ..." Typos should be avoided.
(5) When you mention for the first time such abbreviation as SGTW, you should also spell it.
(6) In methodology, there should be mentioned only in short paragraphs which instrument and specific settings (column, temperature, ramp, etc.) was used. Also a paragraph that explains why you selected these instruments because there are many options such as FT ICR, GCMS, GCFID, Fluorescence, etc.
(7) Try to argue for the selection of your catalyst in the results and also explain the experimental matrix and design of your experimental work using statistics. See the example from "Production and characterization of bio-oil from fluidized bed pyrolysis of olive stones, pinewood, and torrefied feedstock"
(8) It is absolutely mandatory that you also clearly state which conditions of bio-oil production you are going to study and also as I said before try to use statistics to support your argument.
(9) I generally do not like your results chapter. There are so many publications about the gasification and pyrolysis. It would be great to show all differences to the previous literature and novelty. Changing a feedstock and making gasification is not a novelty, but really showing or discussing a mechanism is of great importance.
(10) Discussion section and comparison to the previous results and maybe summarizing all results in a mathematical or chemical model will be of interest to the reader.
(11) Conclusion is too long and has too many details but does not show the novelty.
Author Response

(The authors gave the same response as above.)

Round 2
Reviewer 1 Report
This manuscript is well revised and it can accepted for publication.
Reviewer 3 Report
The author have not improved their manuscript with the novelty remaining unclear. I believe that there are so many articles in that research area that specifically this publication will be not so attractive to readers.